

# Physical, mechanical and energy characterization of wood pellets obtained from three common tropical species

Carrillo Parra Artemio[1,*], Ngangyo Heya Maginot[2,*], Colín-Urieta Serafín[3], Foroughbakhch Pournavab Rahim[2], Rutiaga Quiñones José Guadalupe[3] and Correa-Méndez Fermín[4]

[1] Instituto de Silvicultura e Industria de la Madera, Universidad Juárez del Estado de Durango, Durango, Mexico
[2] Facultad de Ciencias Biológicas, Universidad Autónoma de Nuevo León, San Nicolás de los Garza, Nuevo León, Mexico
[3] Facultad de Ingeniería en Tecnología de la Madera, Universidad Michoacana de San Nicolás de Hidalgo, Morelia, Michoacan, Mexico
[4] Desarrollo Sustentable, Tecnologías Alternativas, Universidad Intercultural Indígena de Michoacán, Pichátaro, Michoacán, México
* These authors contributed equally to this work.

Corresponding author
Ngangyo Heya Maginot,
nheyamaginot@yahoo.fr,
rahimforo@hotmail.com

## ABSTRACT

**Background.** The need for energy sources with low greenhouse gas emissions and sustainable production encourages the search for alternative biomass sources. However, the use of biomass fuels faces the problem of storage, transport and lower energy densities. Low-density values can negatively affect energy density, leading to an increase in transportation and storage costs. Use of pellets as alternative biomass source is a way to reduce the volume of biomass by densification, which improves their energy quality. They are produced by diverse biomass resources and mainly from wood materials. In all cases, it is important to evaluate the fuel characteristics, to determine their suitability on the heating system and handling properties.

**Methods.** The present study determines and compares data from proximate analysis, calorific values, physical and mechanical properties of wood pellets produced from the common tropical species *Acacia wrightii*, *Ebenopsis ebano* and *Havardia pallens*. Data were obtained from pellets produced from each species chips collected from an experimental plantation and analyzed through ANOVA and Kruskal–Wallis test at 0.05 significance level.

**Results.** The results of diameter, length and length/diameter ratio didn't show statistical differences ($p > 0.05$) among species. *Acacia wrightii* showed the highest density (1.2 g/cm$^3$). Values on weight retained and compression test showed statistical differences ($p = 0.05$) among species. *Havardia pallens* was more resistant to compression strength than *A. wrightii* and *Ebenopsis ebano*. Statistical differences ($p < 0.01$) were also observed for the volatile matter and calorific value. *E. ebano* has the lowest volatile matter (72%), highest calorific value (19.6 MJ/kg) as well as the fixed carbon (21%).

**Discussion.** The pellets of the species studied have a high energy density, which makes them suitable for both commercial and industrial heating applications. A pellet with low compression resistance tends to disintegrate easily, due to moisture adsorption. The percentages obtained for the resistance index were higher than 97.5%, showing that the pellets studied are high-quality biofuels. Proximate analysis values also indicate

good combustion parameters. Pellets of *Acacia wrightii* and *Ebenopsis ebano* are the more favorable raw material sources for energy purposes because of their high density, calorific value, low ash content and they also met majority of the international quality parameters.

## INTRODUCTION

The present society development provides increasing levels of comfort to people, inevitably leading to an increase in energy consumption in all its forms (*Van Duren et al., 2015*) that requires a constant and permanent supply (*Song et al., 2015*). It is estimated that 80% to 85% of the world's energy consumption is obtained directly from fossil fuels (*BP, 2013*) which cause greenhouse gas emissions, global warming; in addition; they are limited in supply and they will eventually be depleted. Therefore, it is important to develop new energy policies, aimed at reducing the rate of energy consumption and the environmental impact associated with the use of fossil fuels.

Biomass is a clean source of energy whose use implies a reduction in the energy dependence of fossil fuels (*Antolín, 2006*). Thus, biomass energy is a promising alternative to such limited fossil fuel reserves as coal, oil and gas (*Zhao et al., 2012*) since the natural ecosystems produce more than 230 billion tons of biomass each year, of which only a quarter (24%) is used to satisfy basic needs and industrial production, leaving 76% of the total biomass existing, which can therefore become a living "green" carbon source to supply or partially replace the "black" fossil fuels currently supporting the industrial economies (*ETC Group, 2010*). However, one of the challenges facing the energy industry is how to store the large quantities of biomass fuel required for thermal power plants (*Craven et al., 2015*). Moreover, biomasses are scattered resources with lower energy densities (*Hu et al., 2014*), and to be practical in large-scale applications, they must be first pretreated by grinding, drying and compressing (*Chen, Peng & Bi, 2015*), so that they are dry and dense with a higher energy density.

Densification then appears as a way of producing solid biofuels, easily transportable, manageable and storable, with optimum commercial quality. Densified biomass fuels such as pellets are preferred as they provide better economic viability for transport, storage and handling than other biofuels (*Tauro et al., 2018*). According to *Patzek & Pimentel (2007)*, they are easy to process, transport over long distances, and are relatively safe. Also, wood pellets are an efficient source of biomass energy, which is important, as fossil fuels contribute dramatically to $CO_2$ emissions (*Thomson & Liddell, 2015*), whereas pellets burn cleanly and thus create less air polluting emissions, as explained by *Kowollik (2014)* with the concept of neutral carbon, compared to other combustion heating energy sources. Many scientists and organizations believe that if efforts to develop renewable energy continue, by 2050 renewable energy will provide about 30% of the world's demanded energy and

a significant amount of this energy could come from wood pellets (*Guo, Song & Buhain, 2015*), which are less expensive than fossil fuels, such as oil, liquefied petroleum gas, and electric powered systems, particularly as wood pellets have higher energy content than oil (*Thomson & Liddell, 2015*). In addition, producing wood pellets is very cost effective since the raw materials are relatively cheap and mills can operate automatically needing only a few employees (*Lu & Rice, 2010*). These wood pellets' high availability and the low price of raw materials make their cost more stable, which is especially positive as prices of fossil fuels fluctuate widely (*Roh, 2016*).

Thus, wood as a primary energy source responds to available evidence and to a need for energy; this is especially relevant for a time of deep economic crisis, which has forced many to rethink future strategies (*Brian Vad, Lund & Karlsson, 2011*). In this way, the use of wood pellets is a sustainable energy alternative (*Mola-Yudego, Selkimaki & Gonzalez-Olabarria, 2014*; *Sgarbossa et al., 2015*) that represents a positive globalization of wealth and local employment generation. This has resulted in a soaring demand for wood pellets in Europe and North America (*Heinimo & Junginger, 2009*) so that they are produced by diverse biomass resources, such as wood waste, energy forest and grape marc (*Cespi et al., 2014*; *Dwivedi et al., 2014*). Therefore, the pelletizing can be considered as an option to counteract the problem of excess waste normally generated in agro industrial and forestry activities (*IRENA, 2013*).

In tropical conditions, many agricultural and forestry crops are developed, generating a large amount of lignocellulosic waste (*Ulloa et al., 2004*) that could be used as fuel or energy source (*Sekyere et al., 2004*) through pelletization. However, before these woody pellets can be used, it is essential to first evaluate their fuel characteristics, taking as reference some standards, to ensure their uniformity, reducing market barriers and creating a product flow in which these biofuels can be traded between producers and users regardless of countries or regions (*Cabral et al., 2012*). This is directly related to the physical, mechanical and chemical properties that determine the quality of densified biomass during transportation and storage, as well as their energy capacity. Thus, in this study, three common tropical species are tested and characterized, to compare and determine the suitability of their pellets according to the international standards and end-user's requirements based on the heating system and handling properties.

## MATERIALS & METHODS

### Origin of raw materials and pellets production

Four trees from each of the species Acacia wrightii, Ebenopsis ebano and Havardia pallens were cut from an experimental plantation established in Northeast Mexico (*Ngangyo-Heya et al., 2016*). The material was chipped, and then milled into a particle length lower than 4 mm. The pellets were produced in a press with compression channel length of 8 mm and channel diameter of 6 mm, without adding binder-additives to obtain pellet production of 400 kg/h. The pellets were cooled and left in plastic bags at laboratory conditions for the physical and chemical tests.

## Physical properties

The pelletizing press and wood particles characteristics affect pellets' physical properties such as length, diameter and density. Pellet diameter is the result of the die dimension, and pellet length from the distance between plate and knife placed down the dish; however, particle density is related to pelletizing conditions and wood particles characteristics. The pellets diameter and length was measured for 50 samples of each species with a caliper, and the particle density was determined by the ratio of mass to volume according to Eq. (1). All values were the average of 50 samples of each species.

$$D = m/v \tag{1}$$

where, $D$ = Particle density (g/cm$^3$), m = Mass of pellet (g), V = Volume of pellet (cm$^3$).

## Mechanical properties

Compression resistance at diametrical load was determined for 20 samples of each species, using a universal testing machine (Instron 300Dx; Instron, Norwood, MA, USA), the pellet was placed between two flats and parallel platens, and an increasing load was applied at the constant velocity of 2 mm/min until the pellet failed by cracking or braking according to the test established by *Nielsen, Holm & Felby (2009)*.

Impact resistance also known as "drop resistance" or "shattering resistance" was used to determine the safe height of pellet production (*Kaliyan & Morey, 2009*; *Pietsch, 2008*). The impact resistance index (IRI) was obtained from the total number of pellets pieces produced after dropping each of the 20 pellets per species four times from 1.8 m height. The data was calculated according to Eq. (2), developed by *Richards (1990)*.

$$IRI = 100 \times N/n \tag{2}$$

Where: $IRI$ = impact resistance index, $N$ = number of drops, $n$ = total number of pieces after the four drops.

The retained weight percentage was determined from the weight of the total number of pellets pieces produced from the four drops divided by the initial weight of the pellet multiplied by 100 according to Eq. (3).

$$RW = Wnp/WN \tag{3}$$

where: $RW$ = retained weight (%), $Wnp$ = weight of the total pieces produced after four drops, $WN$ = weight of the initial piece of pellet.

## Proximate analysis and energy production

Moisture content (%), volatile matter (%), and ash content (%) were determined according to the standards *Spanish Association for Standardization (UNE)(2010a)*, *Spanish Association for Standardization (UNE)(2010b)*, *Spanish Association for Standardization (UNE)(2009)*, respectively. Fixed carbon content was calculated by subtracting from the sum of the volatile matter, moisture and ash content from 100. Gross calorific value of pellets was calculated according to Eq. (4) established by *Parikh, Channiwala & Ghosal (2005)*.

$$GCV = 0.3536FC + 0.1559VM - 0.0078A \tag{4}$$

Where, $GCV$ = Gross calorific value (KJ/kg), $FC$ = Fixed carbon (%), $VM$ = Volatile matter (%), $A$ = Ash (%).

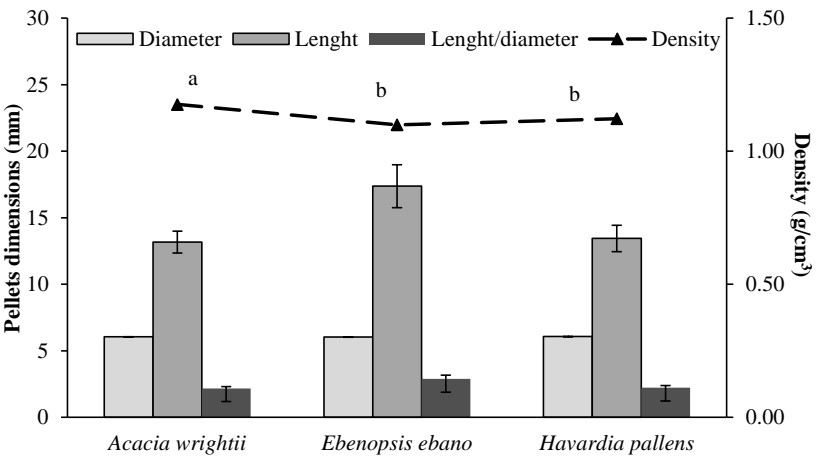

**Figure 1** **Average and standard error of length, diameter, ratio length/diameter and particle density of wood pellets produced from three common tropical species.** Density of species with the same letters are statistically similar ($p < 0.05$) according to Tukey's honestly significant difference test.

## Statistical analysis

The data means and standard error values for the properties of the pellets produced from the three species studied were determined, normality for all variables was corroborated by Shapiro–Wilk. Data in percentage were transformed using the arc sine square root function to develop comparison tests. Variables showing normal distribution were analyzed using one-way analysis of variance (ANOVA) with a random arrangement. Comparisons with statistical differences ($p < 0.05$) between species, Tukey's honestly significant difference (HSD) tests were developed, this test consider statistically significant at $p < 0.05$ for all pair-wise comparisons (*Steel & Torrie, 1960*). For variables non-normally distributed, comparisons among species were developed with Kruskal–Wallis test. All statistical analyses were performed using the free R software, version 3.2.2 R (*Bolker, 2012*).

## RESULTS

### Physical properties

The average and standard error of length, diameter and density of pellets produced with the three tropical species studied are shown in Fig. 1. Pellets density values showed statistical differences ($p < 0.05$) among species (Table 1). The density showed two statistical groups: (a) with pellets of *Acacia wrightii*, which were the denser pellets (1.18 g/cm$^3$), and (b) constituted of pellets obtained from *Ebenopsis ebano* and *Havardia pallens* that were statistically similar, with values of 1.10 and 1.12 g/cm$^3$, respectively.

As for the dimensions, the pellets of the three species have similar diameters: *A. wrightii* (6.05 ± 0.01 mm), *E. ebano* (6.03 ± 0.01 mm) and *H. pallens* (6.07 ± 0.01 mm), while for the length, the *E. ebano* pellets (17.37 ± 1.61 mm) were longer than those of *A. wrightii* (13.17 ± 0.82 mm) and H. *pallens* (13.44 ± 0.99 mm). The ratio length/diameter was 2.89 ± 0.27, 2.22 ± 0.17 and 2.17 ± 0.14 for pellets of *E. ebano*, *H. pallens* and *A. wrightii*, respectively.

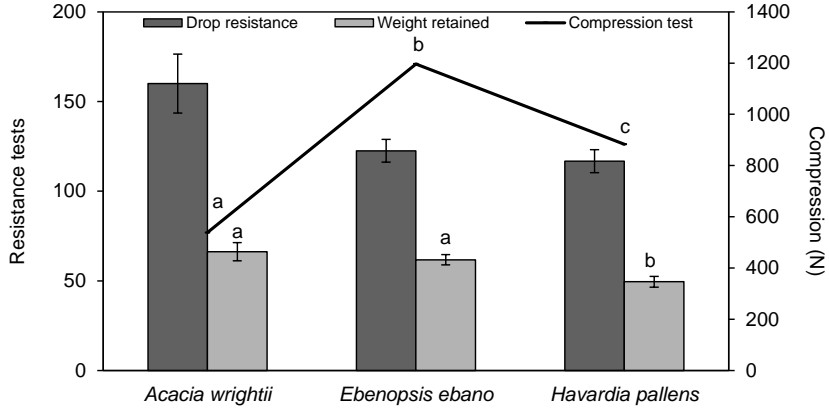

**Figure 2 Average and standard error of drop resistance, weight retained and compression resistance of wood pellets produced from three common tropical species.** Weight retained and compression resistance values of species with the same letters are statistically similar ($p < 0.05$) according to Kruskal test.

**Table 1 Shapiro–Wilk and Kruskal–Wallis tests of the physical properties of pellets elaborated from chips of three common tropical species.**

| Physical property | Shapiro–Wilk test | | Kruskal–Wallis test | |
|---|---|---|---|---|
| | Statistic | p-value | chi-squared | p-value |
| Diameter | 0.79371 | 2.90E–13 | 2.5197 | 0.2837 |
| Length | 0.88039 | 1.19E–09 | 1.4084 | 0.4945 |
| Ratio Length/Diameter | 0.90761 | 3.64E–08 | 1.4516 | 0.4839 |
| Density | 0.94534 | 1.37E–05 | **9.1343** | **0.0104** |

**Notes.**
Value highlighted bold indicated statistical differences ($p < 0.05$) among species.

**Table 2 Shapiro–Wilk and Kruskal–Wallis tests of the mechanical properties of pellets elaborated from chips of three common tropical species.**

| Mechanical property | Shapiro–Wilk test | | Kruskal–Wallis test | |
|---|---|---|---|---|
| | Statistic | p-value | chi-squared | p-value |
| Compression | 0.95677 | 3.28E–02 | **14.868** | **0.0005909** |
| Drop resistance | 0.69085 | 5.94E–10 | 3.9705 | 0.1373 |
| Weigth retained | 0.95128 | 1.79E–02 | **7.7059** | **0.02122** |

**Notes.**
Bold data shows the variable with statistical differences ($p < 0.05$) among species.

## Mechanical properties

Compression resistance values showed statistical differences ($p < 0.05$) among species (Table 2). The bonds between pellet particles produced from *Acacia wrightii* wood chips were stronger than those from *E. ebano*. *H. pallens* has the weakest particle bond (Fig. 2).

The drop resistance index values for the three species ranges between 117 to 160. *A. wrightii* produced the most resistant pellets, while *H. pallens* produced less resistant pellets. The registered values did not show statistical differences ($p > 0.05$) among species
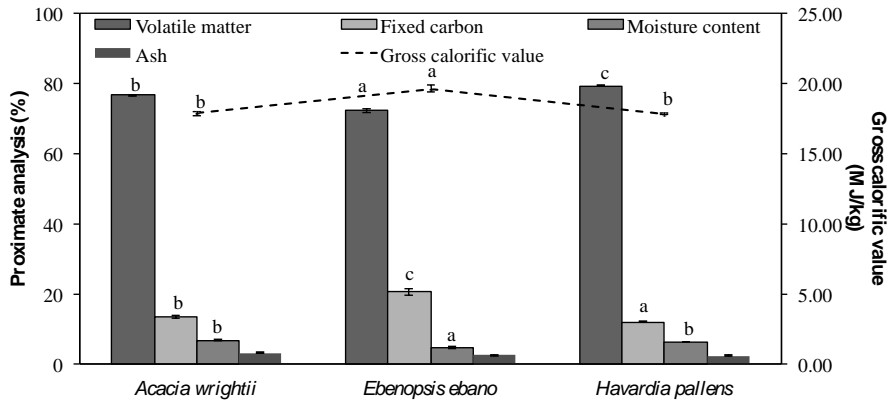

**Figure 3** **Proximate analysis and energy values of wood pellets from three tropical species.** Moisture content, volatile material and gross calorific values of species with the same letters are statistically similar ($p < 0.05$) according to Tukey's honestly significant difference test, and by fixed carbon values according to Kruskal test.

**Table 3** **Shapiro–Wilk and Kruskal–Wallis tests of the proximate analysis and energy values of pellets elaborated from chips of three common tropical species.**

| Proximate analysis and energy values | Shapiro–Wilk test | | Kruskal–Wallis test | | Anova test | |
|---|---|---|---|---|---|---|
| | Statistic | *p*-value | chi-squared | *p*-value | *F* value | *p*-value |
| Moisture content | 0.92759 | **0.4588** | | | **32.40** | **0.000609** |
| Volatile matter | **0.93752** | **0.556** | – | – | **59.12** | **0.000113** |
| Ash | **0.96381** | **0.8373** | – | – | 1.764 | 0.25 |
| Fixed carbon | 0.82073 | 0.03512 | **7.2** | **0.02732** | – | – |
| Gross calorific value | **0.83748** | **0.05415** | – | – | **23.14** | **0.00151** |

**Notes.**
Bold data shows the variable with statistical differences ($p < 0.05$) among species.

(Table 2). Weight retained values showed statistical differences ($p < 0.05$) among species (Table 2). Pellets of *A. wrightii* and *E. ebano* were in the same statistical group "a" with values of 66.27 and 61.74, respectively, different from those of *Havardia pallens* in the statistical group "b", with the value of 49.49 (Fig. 2).

## Proximate analysis and energy production

Moisture content, volatile matter, fixed carbon and gross calorific value showed statistical differences ($p < 0.005$) among species, while ash showed similar values among species (Table 3). The moisture content of all the pellets produced from the common tropical species tested was lower than 7%, with *E. ebano* (4.62 $\pm$ 0.23%), *H. pallens* (6.18 $\pm$ 0.03%) and *A. wrightii* (6.74 $\pm$ 0.15%) as presented in Fig. 3. Volatile matters oscillated from 72.25 to 79.38%, values corresponding to pellets of *E. ebano* and *H. pallens*, respectively. The ash content ranged between 2.41 to 3.22%, being the smallest value obtained for pellets of *H. pallens* and the highest value for pellets of *A. wrightii*. Fixed carbon varied significantly, with values ranging between 12 to 21%, thereby forming three statistical groups: "a" *H. pallens* (12.03 $\pm$ 0.11%), "b" *A. wrightii* (13.44 $\pm$ 0.51%) and "c" *E. ebano* (20.61 $\pm$ 1.01%).

Calorific values found in this research were higher than 17.8 MJ/kg, and is the highest value obtained from pellets of *E. ebano* (19.64 MJ/kg).

## DISCUSSION

### Physical properties

The values of pellets density obtained in this work are similar to 1.12–1.3 g/cm$^3$ reported in The Pellets Handbook by *Thek & Obernberger (2012)*. Comparing wood density average, *Rodriguez et al. (2016)* found that wood pellets density of *A. wrightii*, *E. ebano* and *H. pallens* increased 30%, 25% and 93%, respectively, which confirm that pelletization is a good process to increase the density even to denser species. The bulk density of the input material is an important factor in pelleting as the mills are fed by volume rather than weight (*Filbakk et al., 2011a*). The consideration for this property is a good estimator of pellet quality for fuel applications, as it equates to more energy per unit volume, and means greater economy in fuel use, transportation and storage space (*Rollinson & Williams, 2016*). Low density values can negatively affect the energy density causing an increase in transportation and storage costs. According to *Obernberger & Thek (2006)*, the pellets of high energy density (18–20 MJ kg$^{-1}$) are suitable for both commercial and industrial heating applications.

Comparisons between physical properties of pellets studied against values stated by standards CEN/TS 14961 (CEN/TS EN 14961-2, 2012), SS 18 71 20 and CTI R04/05 showed that they met the standards (*Duca et al., 2014*). According to the CEN-EN 14961-1, pellets from the three species were "D 06" with a diameter of 6 mm and length between 3.15 to 40 mm (*CEN/TC 335-Solid Biofuels, 2005*). All pellets of the studied species are suitable for combustion in boilers with pneumatic feeding systems because their lengths were small enough to prevent a blockage in the mechanism. Also, the ratio of length and diameter was lower than maximum of five stated by *Obernberger & Thek (2006)* and *ÖNORM (2000)*.

### Mechanical properties

The compression resistance values of *Acacia wrightii* were within the range (295 to 692 N) reported by *Tenorio et al. (2014)* and *Pampuro et al. (2017)*, while compression resistances of pellets of *E. ebano* and *H. pallens* were higher than the values reported. For wooden pellets, the resistance to change from its original appearance is very important, since it indicates how well they can resist external forces after a sustained period of use. This property is important in the wood pellet industry and trade (*Oveisi-Fordiie, 2011*). A pellet with low compression resistance is usually associated with problems such as difficulty in storage and shipping as well as health and environmental issues. This is because such pellet has the tendency to disintegrate easily due to moisture adsorption, fall or friction as reported by *Temmerman et al. (2006)*. Thus, measuring this parameter for pellets indicates their market values.

The values of the impact resistance index were higher than the ratio of 33 to 50% reported by *Forero-Nuñez, Jochum & Sierra (2015)*. Pellets with a percentage higher than 97.5% as defined by *ASABE Standards (2006)* are considered a high quality biofuel because

the particles have good adhesion forces that allow pellets withstand transportation stress before reaching to the end users.

## Proximate analysis and energy production

The moisture content of about 7% for all the studied species is in conformity with *Koppejan & Van Loo (2012)*, who stipulated that moisture content of quality pellets should be lower than 15%. Moisture content values place the studied pellets as super premium ($\leq 8\%$), according to US standard which has other three lower grades, i.e., premium ($\leq 8\%$), standard ($\leq 10\%$) and utility ($\leq 10\%$) (*Tumuluru et al., 2010*). Moisture content is a property that should be considered with caution, since water has a crucial role in the pelletizing process (*Samuelsson et al., 2009*). A number of studies on wooden pellets showed a positive correlation between MC and pellet durability (*Whittaker & Shield, 2017*), being this, one of the most important physical characteristic of pellets. Higher MCs can reduce friction by lubricating the biomass (*Nielsen, Holm & Felby, 2009*), and increase the extent at which pellets 'relax' after formation thereby leading to a decrease in durability (*Adapa, Tabil & Schoenau, 2011*). Water is not compressible, however, limiting the final density of the pellet (*Carone, Pantaleo & Pellerano, 2011*). When moisture content is at the level of 8.62%, the maximum durability of 96.7% can be reached (*Colley et al., 2006*). With MC of 8–15%, there is an increase in durability in Norway spruce and Scots pine (*Lehtikangas, 2001*). *Filbakk et al. (2011b)* also found a positive correlation in durability ($r2 = 0.62$) with MC of 7–12% in Scots pine. Tulip wood pellets showed the highest durability at a moisture content of 13% (*Lee et al., 2013*). Across a range of biomass types including wood and straw, the optimum MC for pellet durability was between 6.5 and 10.8% (*Miranda et al., 2015*).

The volatile matters range (72 to 79%) is in agreement with the results of *Arranz et al. (2015)*, *Koppejan & Van Loo (2012)*, *Tenorio et al. (2014)*, and are lower than 82.8% reported by *Chen, Peng & Bi (2015)* for commercial pellets. The amount of volatile matters influences the behavior during the combustion of solid fuels (*Tauro et al., 2018*) such that when volatile matters are high, the biomass is considered a suitable fuel for thermal conversion (*Olsson & Kjällstrand, 2004*; *Holt, Blodgett & Nakayama, 2006*). *Kataki & Konwer (2002)* additionally indicated that high levels of volatile matters produce a fast burning, a disadvantage to fuels.

Fixed carbon varied from 12 to 21%, similar results were reported by *Chen, Peng & Bi (2015)* and *Arranz et al. (2015)*. Fixed carbon has been reported to influence the gross calorific value (*Tenorio et al., 2014*). Also, in relation to the potential of energy production, this property is the most valuable parameter, since raw materials with high fixed carbon have higher heating values (*Forero-Nuñez, Jochum & Sierra, 2015*).

The ash content ranged between 2.41 to 3.22%, which is promising for the species studied. Pellets with low ash contents are suitable for thermal conversion because they cause low ash accumulation, slagging or corrosion in the boilers (*Obernberger & Thek, 2006*; *Rhén et al., 2007*). Ashes reduce the quality of pellets, increase the emission of particles to the environment and reduce the heat value of biomass (*Tumuluru et al., 2010*). According to *Uribe (1986)*, the higher the ash content in a solid fuel, the lower will be the heat obtained, causing problems with the handling and management of large quantities of ash

produced. High ash content feedstock may also result in increase in maintenance cost for both household and industries users. The relatively small amount of ash indicates small ash forming elements, allowing the pellets to be used for industrial heating requirements, where problems associated with slagging, fouling and sintering formation are major concerns.

Gross calorific values found in this research were higher than 17.8 MJ/kg, that are similar to the values reported by *Telmo & Lousada (2011)*, indicating that pellet of the studied species are suitable to be used as feedstock (*Laxamana, 1984*; *San Luis, Briones & Estudillo, 1984*). Also, these values are within the minimum requirements of *DIN 51731 (1996)* for solid fuel, for industrial heating processes. High gross calorific values allow the biofuel to produce a high amount of energy within low fuel volume (energy density) (*Atuesta-Boada & Sierra-Vargas, 2015*).

## CONCLUSIONS

The wooden chips of the common tropical species *A. wrightii*, *E. ebano* and *H. pallens* from experimental plantations are suitable to produce pellets that meet the international quality parameters. Pellets' physical parameters values such as length/diameter ratio and density indicate a well bonding mechanism. Resistance, compression and weight retained values of the three species guarantee that the pellets will produce low levels of fines during transportation. Proximate analysis values indicate good combustion parameters for the species. Pellets of these species are classified as M10 (moisture content lower than 10%) and A0.5 (ash content lower than 0.5%). Gross calorific values from all three species were higher than 17.8 MJ/kg. From the values of the wood pellets studied, *A. wrightii* and *E. ebano* are the more favorable raw materials sources for energy purposes because of their high density, gross calorific value, and low ash content, which met the majority of international quality parameters.

### Funding
This work was supported by the Energy Sustainability Fund through the project SENER CONACYT 2014 (No 246911): Cluster of Solid Biofuels for Thermal and Electric Generation. The funders had no role in study design, data collection and analysis, decision to publish, or preparation of the manuscript.

### Grant Disclosures
The following grant information was disclosed by the authors:
Energy Sustainability Fund through the project SENER CONACYT 2014: No 246911.

### Competing Interests
The authors declare there are no competing interests.

## Author Contributions

- Carrillo Parra Artemio conceived and designed the experiments, performed the experiments, analyzed the data, contributed reagents/materials/analysis tools, prepared figures and/or tables, authored or reviewed drafts of the paper, approved the final draft.
- Ngangyo Heya Maginot conceived and designed the experiments, performed the experiments, analyzed the data, prepared figures and/or tables, authored or reviewed drafts of the paper, approved the final draft.
- Colín-Urieta Serafín performed the experiments, authored or reviewed drafts of the paper, approved the final draft.
- Foroughbakhch Pournavab Rahim analyzed the data, authored or reviewed drafts of the paper, approved the final draft.
- Rutiaga Quiñones José Guadalupe conceived and designed the experiments, contributed reagents/materials/analysis tools, authored or reviewed drafts of the paper, approved the final draft.
- Correa-Méndez Fermín performed the experiments, prepared figures and/or tables, authored or reviewed drafts of the paper, approved the final draft.

## Data Availability

The raw data are provided in Data S1.

## Supplemental Information

Supplemental information for this article can be found online at http://dx.doi.org/10.7717/peerj.5504#supplemental-information.

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
