# Peer review of "Physical, mechanical and energy characterization of wood pellets obtained from three common tropical species"

_PeerJ, doi:10.7717/peerj.5504_

## Round 0.1 · original submission · Major Revisions

Three reviewers have reviewed the manuscript and have agreed that it may be suitable for publication after addressing the comments they raise.

Having carefully read the manuscript myself, I have a few additional comments for your considerations:

1) the current manuscript contains several typos and issues with the English expression (e.g., "Kruskal-Walliss" or "Kurskal-Walliss" instead of "Kruskal-Wallis"; "Lenght" instead of "Length" in some of the charts). One of the reviewers has provided detailed edits but some issues remain. Please make sure that the text is carefully proofread before resubmission.

2) Figures 1-3 provide the results of pair-wise comparisons also for non-normally distributed variables (as determined by the Shapiro-Wilk test), but the methodology section is not explicit about which test was used for this analysis. Please clarify.

3) Please revise the captions of Figures 1-3 to reflect the fact that "Species with the same letters are not statistically different (p>0.05)" and to clarify that this follows from the pair-wise comparison tests.

4) There seems to be an issue in Table 3 for the variable "Moisture content". Why was it tested with the Kruskal-Wallis test when the Shapiro-Wilk test does not reject normal distribution? This is not consistent with the other variables. Also, the numbers reported for the chi-squared test and p-value are identical to those reported for "Fixed carbon" which may suggest some editing issue in the table. Please double check.

Reviewer 1 ·

Basic reporting

The article is well written and informative. literature and references are sufficient, short and specific.Data are seems to be from smaller samples.

Experimental design

Its original primary research and have scope for better utilisation of wood pellets with high calorific values. The correlation between different parameters also well established .

Validity of the findings

Novelty of finding is reconfirming density and IRI of wood of selected tropical species with heating value. the pellet densities are 1.10 to 1.18 g/cm3 in three species but what is basic density of these woods is not mentioned and not a relationship with basic density of wood with their pellet density is unfolded.
Is there any method or formulation to increase the pellet density of inferior or lighter wood biomass/ chips to be used as pellet for better use.
The wood billets or chips from high density or specific gravity has high calorific value/ heating value than why we need to make their pellets, as these can be directly used in improved thermal heaters or wood gasifier. Pelleting has major role where wood not have good calorific value and easily combustible to mix its particle with charcoal or coal and other additives to have briquettes for better use.
I think pellets and briquette are same terms.

Additional comments

The article is good and concise.

·

Basic reporting

No comments

Experimental design

No comments

Validity of the findings

No comments

Additional comments

The manuscript is suitable for publications. The minor corrections are made in the manuscript which may be incorporated before publication. There should be a title in tables and figures which are lacking. The corrected manuscript is sent via mail.

·

Basic reporting

Clear and unambiguous, professional English used throughout.

Experimental design

Original primary research within Aims and Scope of the journal.

Validity of the findings

Conclusion are well stated, linked to original research question & limited to supporting results.

Additional comments

The paper is interesting and actual. It is a subject that has direct application.

In my opinion the authors didn’t presented a very well structured abstract and didn’t showed properly the importance that the article may have.

Would be important if the authors could present some examples for the potential uses of this.

This paper must be revised to include a short history about the usage of this type of solutions and the impact that this may have.

The following suggestions are intended to be seen as a help to the authors in order to improve the final version of the paper to be ready for publication.

1. Introduction can be improved to better explain the importance of the subject.
2. Authors should review more the references used. It is possible to find more references about this subject or related references and many important are missing.
3. Tables and figures must be better explained in plain text or modified to be more understandable.
4. Authors should try not to overkill references using several in the same paragraph without a clear understanding of the interest of each one.

---

## Round 0.2 · Minor Revisions

Most of the comments previously raised were adequately addressed by the authors. There remain however a few minor additional issues - primarily with the English expression - that need to be addressed before the paper is suitable for publication:

L22: Replace "Encourage" with "Encourages"
L28: Replace "on" with "for"
L33: Replace "each species chips" with "chips of each species"
L41: Remove "the"
L56: replace "requiring" with "require"
L59: replace "they will be depleted one day" with "they will eventually be depleted"
L60 is incomplete. Do you mean "aimed at REDUCING the rate of ..."?
L64 "the earth". Do you mean natural ecosystems? Terrestrial ecosystems? Forest ecosystems? Can you cite a source for the numbers you present in this line?
L80-81: clarify what you mean by "while pellets burn cleanly and thus create less air polluting emissions compared to other combustion heating energy sources". Surely there are CO2 emissions from burning pellets as well?
L105-106 "taking as reference of some standards": not clear what you mean here, please rephrase
L163: add the acronym "(ANOVA)" after "one-way analysis of variance"
L166: replace "comparisions" with "comparisons"
L174: replace "more dense" with "denser"
L175: rephrase as "constituted of pellets obtained from Ebenopsis ebano and Havardia pallens that were statistically similar..."
L211: replace "ood" with "wood"
L231: replace "while pellets" with "while compression resistances of pellets"
L244: remove "to"
L250: "standard" and "utility" are both reported as <10%. Please correct
L269: remove "as"
L284: remove "and as as result"
L285: replace "into" with "in"
Table 1: replace "Lenght" with "Length"
Table 2: replace "Kruskal" with "Kruskal-Wallis"

---

## Round 0.3 · accepted · Accept

Please carefully proofread the final manuscript to ensure that some remaining typos are dealt with during the production phase.

#